# The impact of internet use in the digital era on public political trust in China—An empirical study based on CGSS 2021 data

Yanan Wen[1☯], Xue Sun[2☯], Lu Fang [3]*

1 School of Public Administration and Law, Anhui University of Technology, Ma'anshan, Anhui, China,
2 School of Sociology and Political Science, Anhui University, Hefei, Anhui, China, 3 School of Public Administration, University of Electronic Science and Technology of China, Chengdu, Sichuan, China

☯ These authors contributed equally to this work.
* fnglu1017@sina.com

## Abstract

### Background

The system of the congresses of the Chinese people is the fundamental political system established by the Chinese people under the leadership of the Communist Party of China (CPC), suited to China's national conditions. As China's highest national authority organ, the National People's Congress (NPC) plays an important role in China's development. The trust of the Chinese public for the NPC objectively reflects the level of trust by the Chinese public and their confidence in the people's congress system.

### Methods

Based on Chinese general social survey (CGSS) 2021 data and using SPSS22 software, this paper examines the impact of public Internet use on political trust through binary logistic analysis and ordinal regression analysis methods.

### Results and Conclusions

Research showed that the use of the Internet and its use frequency does not affect public's responsiveness to political trust questions, but the use of the Internet affects the political trust level of the public. People who do not use the Internet at all and those who use the Internet daily are significantly different, and the former hold a higher political trust level than the latter. Moreover, the level of trust of people using the Internet several times per week is markedly lower than that of daily Internet users; no significant differences were found in the level of trust of people using the Internet several times a year or less, several times a month, and daily.

**Data availability statement:** The data used in this paper are true, scientific, objective and publicly available, and the data used in this paper are from the CGSS 2021, which can be accessed at the website http://www.cnsda.org/index.php?r=projects/view&id=65635422, and its project number is 65635422. This study did not require ethical review committee review and consent, nor did it require informed consent, as the data were obtained from the publicly available CGSS database. Additionally, the author has uploaded the dataset used in this paper (SAV format) to the figshare platform, with the DOI: 10.6084/m9.figshare.29755613; Considering that SAV format requires SPSS software to open, the author has also saved the SAV dataset in xlsx format and uploaded it to the figshare platform, with the DOI: 10.6084/m9.figshare.30674195.

**Funding:** This study was funded by the Anhui Provincial Department of Education (http://jyt.ah.gov.cn/) Key Project in Humanities and Social Sciences [No. SK2020A0177] awarded to YW (Yanan Wen). and the Graduate Research Innovation Project [SZCXZZS202408] , School of Sociology and Political Science, Anhui University(https://szxy.ahu.edu.cn/), awarded to XS (Xue Sun). The funders had no role in study design, data collection and analysis, decision to publish, or preparation of the manuscript.

**Competing interests:** The authors have declared that no competing interests exist.

## Introduction

Political trust reflects the basic evaluation orientation of the public towards the government, based on the extent to which the government operates in accordance with the public expectations [1]. Political trust is necessary to ensure the stability of democratic political systems [2]. Trust always has an object, which can be an institution, a group, a person, an organisation, or a specific action [3]. Trust originates from one party's conviction that the other party is reliable and will act impartially while safeguarding the interests of the trusting party [3]. Specifically, in politics, these interests may include good policy, the goals of peace, sound economic management, and the welfare of the citizens [3]. The People's Congress system in China is an institutional arrangement that ensures the Chinese people can exercise their political rights in accordance with the law and participate in the management of state affairs. Over the past few decades, the NPC of China has played an important role in China's policy agenda setting, policy formulation, and policy supervision [4].

The CPC attaches great importance to the persistence and improvement of the People's Congress system as well as the people's management of state affairs through the People's Congress. Xi Jinping, General Secretary of the Central Committee of the CPC, pointed out that "the People's Congress system insists that all state power belongs to the people, ensures the people's status as masters of the country to the greatest extent", "the People's Congress system correctly handles a series of major political relationships related to the future and destiny of the country, realizes the unified and efficient organization of various undertakings in the country, safeguards national unity and ethnic unity, and effectively ensures the stability and orderliness of national political life" [5]. The report of the 20th National Congress of the CPC pointed out that the people's exercise of state power is supported and guaranteed through the People's Congress; moreover, it is ensured that the people's congresses at all levels are democratically elected, accountable to the people, and subject to supervision by the people [6].

The 53rd "Statistical Report on China's Internet Development Situation", released by the China Internet Network Information Center (CNNIC), an official organisation in China, showed that, as of December 2023, the number of Internet users in China has reached 1.092 billion, which includes 24.8 million new Internet users compared with the same period in 2022; China's Internet penetration rate has reached 77.5% [7]. China's Internet has been fully popularized and may replace or at least change the existing information environment and communication methods [8]. The Internet provides convenience for the dissemination of various types of information, which affects public opinion and evokes public awareness of rights [8], and will also lead to less public submission to public sector authority [9]. Therefore, the popularity and application of the Internet may impact the public's political trust. Against the backdrop of the rapid development of Internet technology, examining the correlation between Internet use and the trust of the Chinese public in the NPC is greatly important for a deeper understanding of this organizational arrangement with Chinese characteristics. Based on this, this paper uses data from the CGSS 2021 to specifically analyze

the influence of Internet use and use intensity by the Chinese public on trust in the NPC. At the same time, an interesting phenomenon was also discovered, namely that among the total sample of 8,148 respondents, only 2,613 responded to relevant questions, while nearly 2/3 of all respondents did not respond this question. This necessitates that this article pay attention to questions about public responsiveness to political trust. Therefore, this paper examines the impact of internet use on public political trust, encompassing both its influence on responsiveness to political trust questions and its effect on the overall level of public political trust.

This study advances the literature by investigating the influence of Internet use on public responses to questions about political trust, extending beyond prior research that primarily focused on its impact on levels of political trust. At the same time, most of the existing research focuses on the effect of Internet use on the level of public political trust in Western-style democracies, and this research also reveals how Internet use affects the level of public political trust in authoritarian countries.

## Theoretical basis and literature review

### The conceptual connotation of political trust

Political trust exerts a critical impact on citizens' political behavior, and affects the legitimacy of a democratic system [10–12]. The competing theoretical perspectives of culture and institutions represent the primary frameworks for explaining political trust [13]. Based on the cultural theory perspective, political trust is based on attitudes and values [14]. It is an extension of trust between people, that is learned at an early age and is then transferred to political institutions much later [14]. In contrast to the cultural perspective, institutional theory explains political trust from an internal standpoint, contending that institutional trust is essentially the utility the public anticipates when institutions function effectively [13]. From an institutional perspective, political trust has a rational foundation, grounded in and shaped by individuals' evaluations of institutional performance [13,14].

In general, political trust is a multi-dimensional open concept that reflects citizens' assessments of the competence and responsibility of political institutions [15]. According to its classification, political trust is usually defined to explain citizens' feelings towards an entire political group, a regime (such as democracy), as well as any attitudes towards specific political institutions or political actors [16]. Political trust can be studied at the micro level of individuals' willingness to trust, or it can be analyzed at the average level of an individual's political trust at the macro level of a country [16].

### Factors affecting political trust

Political trust is influenced by multiple factors, encompassing socioeconomic conditions, institutional frameworks, cultural dimensions, personality traits, and political efficacy metrics [17–19]. However, some studies also believe that institutional factors, especially the economic and political performance of the government, are important determinants of political trust, while the influence of cultural factors is weaker or insignificant [20]. Specifically, in mature democracies, economic satisfaction, government responsiveness, democratic attitudes, political interest, and religious belief have significant positive explanatory power for political trust; on the contrary, post-materialism, political radicalism, corruption indulgence, and income level impede political trust [21]. In other words, in mature democracies, the higher an individual's income level, the lower their level of political trust. At the same time, income level has different effects in different countries. While income level reduces trust in mature democracies, it increases trust in countries of Latin America and Eastern Europe, but has no effect on the countries of the former Soviet Union [21].

Demographic factors such as occupation, education, age, and interpersonal relationships also impact political trust. Political trust is generally higher among those who have worked in the public sector than among those who have not [22]. In general, political trust is higher among the public with higher levels of education [22]. However, this effect is not evident for people with experience in the public health service, employment service and social service sectors; older people

generally trust government agencies more than younger people [22]. Education has different effects on public political trust in different countries. In mature democracies and the countries of Eastern Europe, education has no effect on political trust, whereas in the countries of Latin America and the former Soviet Union, education is negatively related to trust, while publics with higher levels of interpersonal trust generally have higher levels of political trust [21]. In addition, studies have pointed out that exposure to different media has different effects on the political trust of the public. For example, exposure to official media is positively related to political trust, while exposure to personal media and overseas media is negatively related to political trust; exposure to commercial media is not significant [23].

## The impact of Internet use on political trust

With the advent of the digital age, the impact of the Internet on political trust has gradually entered the horizons of scholars. Current research mostly focuses on the influencing factors of political trust such as whether to use the Internet, Internet usage intensity, and Internet usage preferences. In general, the use of the Internet leads to citizens' distrust of political institutions [24]. Both in democratic and non-democratic countries, the use of the Internet exerts huge downward pressure on political trust [24]. However, the damaging effects on trust caused by the Internet are less severe in democratic countries than in non-democratic countries [24]. Research has found that Internet users exhibit lower political trust than non-Internet users [25]. Compared with people who do not use the Internet at all, new Internet users have significantly lower levels of political trust in local government bureaucrats [26]. At the same time, individuals' time spent on the Internet and political trust showed a negative correlation, the more time individuals spend on the Internet, the lower their trust in the government [8,27]. However, this negative effect is moderated by browsing and accessing government websites [28]. In the new information environment created by the Internet, political trust can been strengthened; in terms of Internet censorship, political trust could be strengthened by Internet blockades but it is weakened by violations of user rights; in terms of Internet participation, electronic information and e-consultation have strengthened political trust, but electronic decision-making undermines political trust [29]. In terms of traditional media and the Internet, regulated television and radio have positively impacted trust in upper-level power institutions, while the less regulated Internet has strong positive impact on trust in central government, the impact on trust in local government was negative [25,30,31]. Different information sources lead to different levels of political trust among the public. Information obtained from official channels promotes political trust [9], while information obtained from personal blogs and foreign websites negatively affects political trust. Political elites drive the consumption of information from news sites and promote democratic support and political trust; however, the disintermediated nature of social media and its bottom-up structure negatively impacts trust in political institutions [28]. It is therefore clear that Internet use will have an impact on the political trust of the public, but this may be closely related to the public's Internet usage preferences, intensity, and information sources.

## Research hypotheses

According to relevant CGSS reports [32,33], the 2021 CGSS employed in-home interviews for data collection. This raises the issue of the respondents' trust in the interviewers [32]. Interpersonal trust encompasses multiple dimensions, ranging from trust in relatives, friends, and acquaintances to trust in strangers [34]. Interpersonal trust entails positive expectations regarding the behavior of others, thereby motivating individuals to engage in interactions through sharing, mutual assistance, and cooperation [35], while also enhancing their willingness to disclose personal information [36]. Trust is essentially the willingness to take the risk of information disclosure [37]. Political trust refers to the public's subjective evaluation and attitude toward political institutions and politicians [38]. An article on the official website of the Cyberspace Administration of China classifies political opinions and other personal information as sensitive [39]. Similarly, the European Union's *General Data Protection Regulation* (GDPR) specifies that individuals' political opinions constitute special categories of personal data [40]. Thus, from this vantage point, individuals' responses to questions regarding political trust fall within the scope of personal privacy.

However, Chinese scholars have utilized data from the CGSS 2013 and CGSS 2015 surveys to demonstrate that Internet use exerts a significant negative effect on public trust in strangers [41]. Internet use diminishes respondents' trust in interviewers, thereby reducing the propensity for sharing, mutual assistance, and cooperation between them. Trust fundamentally entails the willingness to assume the risks associated with information disclosure [37]; consequently, owing to the absence of trust between them, respondents are reluctant to respond to questions pertaining to political trust. Drawing on the foregoing analysis, we advance the following two hypotheses.

Hypothesis 1: People who use the Internet are less likely to respond to the question of trust in the NPC of China than people who do not use the Internet.

Hypothesis 2: The higher the frequency of public Internet use, the less likely the public is to respond to the question of trust in the NPC of China.

Formally proposed in 1947, gatekeeper theory emphasizes that channels help to more accurately define how certain "objective" social flows are intertwined with "subjective" psychological and cultural topics [42]. Gatekeepers who were introduced into the field of news communication believed that news communication is subjective and relies on their own experiences, attitudes, and expected value judgments; these are passed from one gatekeeper to another in their communication [43]. Shoemaker suggested that gatekeeping is the process of screening, shaping, and refining a large amount of potential news information into information that is actually disseminated by the news media [44]. Gatekeeping in mass communication can be regarded as the entire process of the construction of the social reality of news media dissemination, rather than just as a series of "in" and "out" decisions [44].

In the modern information society, the power of gatekeepers seems to be weakening. The Internet has subverted the concept of gatekeepers, and the amount and quality of online information are so vast that it is even more necessary for someone to classify it and give it credibility [45]. This development may be due to the collective nature of the Internet that makes it possible for citizens to expose government abuses of power [46]. In addition, the human brain naturally prefers unusual and threatening information, and the network will extend its control capabilities to the nodes of the network [47]. Sources that are outside of the scope of traditional news and lack professional standards will become the birthplace of news alongside news giants [47]. "Breaking news articles" and headline-grabbing are prevalent [47]. According to the theory of frame effects, when the same problem is proposed in different ways, the dominating understanding of decision-making issues and the psychological principles of evaluation results will produce predictable changes in preference [48]. This is the important cognitive bias, wherein individuals exhibit differential responses to a specific choice contingent upon whether it entails a loss or a gain [49]. This framework will affect the own perception of a reporter of an incident [50]. When the media reports a certain incident or phenomenon, different expressions will strongly affect the public position [51]. For example, articles on corruption published in newspapers controlled by authoritarian regimes will reduce the level of public hatred towards corruption, while exposure to trail news will increase the level of public hatred towards corruption [52]. Consequently, the inherent characteristics of the Internet, coupled with human preferences, collectively facilitate the dissemination of rumors and unsubstantiated information online, thereby rendering Internet users more susceptible to exposure to negative and false content compared to non-users. Consequently, we formulate the following as our third hypothesis.

Hypothesis 3: People who use the Internet have lower levels of political trust in the NPC of China than people who do not use the Internet.

Algorithms are a determining factor in today's media technology, and "algorithmic gatekeepers" have been an important feature of digital journalism since at least the inception of Google News in 2002 [53]. Compared to news processed by journalists, 54% of global news users prefer algorithmic forms of news presentation; these news users primarily use smartphones (58%) and are young (64%) [53]. The role of algorithms as gatekeeper is increasing, as algorithms largely determine which content is promoted and which content is hidden [54]. They have become part of many processes that construct social reality [55]. Although algorithms can facilitate user activity, sharing, and collaborative or automatic ranking,

they also employ personalization that is often imperceptible to users [55]. This has further fueled concerns about filter bubbles such as algorithms in search engines and recommendation systems that determine what can be perceived by the public [55]. Therefore, the use of algorithm gatekeepers can lead to individuals receiving personalized but fragmented news [53]. Specifically, algorithmic recommendations may narrow the scope of an individual's exposure to information, isolating the individual from a wider range of information [53]. Clicks, likes, and shares lead to personalized recommendations that support the beliefs of users [53]. Algorithms may lead to greater fragmentation of news sources, i.e., algorithms that steer users away from traditional news organizations and towards niche sources that are more aligned with user preferences [53]. Meanwhile, algorithmic recommendation systems capture users' personal preferences to deliver tailored recommendations, a function enabled by tracking their website visits and content browsing behaviors [56]. Consequently, the more frequently users engage with the Internet, the better these systems can align with individual preferences, thereby increasing the likelihood of fostering echo chambers and filter bubbles.

The Chinese government has established a comprehensive "gatekeeper" system for internet information dissemination. It has introduced and refined a series of laws and regulations, such as *the Cybersecurity Law of the People's Republic of China*, *the Administrative Measures for Internet Information Services*, and *the Administrative Provisions on Internet News Information Services*, to regulate the release and dissemination of online information and foster a healthy internet ecosystem [57]. However, some studies suggest that prolonged exposure to official media or positive news coverage may also lead to negative public attitudes toward the government [58]. A potential explanatory mechanism is that frequent exposure to online propaganda triggers information overload, thereby prompting the public to think more deeply about the information and further compare such information with political realities [58]. Moreover, empirical studies demonstrate that higher frequency of internet use among the public is associated with lower levels of political trust [25,59]. Drawing on the above analysis, Hypothesis 4 is formulated as follows: The higher the frequency of public Internet use, the lower the political trust of the public in China's NPC.

## Data sources and variable design.

### Methods

The paper begins with a descriptive statistical analysis that reports the mean, median, standard deviation, minimum, maximum, valid and missing values of the variables to help the reader have a comprehensive understanding of the data. Secondly, given that the dependent variable, responsiveness to political trust questions, is dichotomous in nature, this study employs binary logistic regression analysis to examine the impact of both "Whether to use the Internet" and "Internet use frequency" on responsiveness to political trust questions. Thirdly, considering that the dependent variable represents the level of political trust, which is an ordinal variable, this study employs ordinal regression to analyze the effects of both "Whether to use the Internet" and "Internet use frequency" on political trust levels. To ensure methodological rigor, this study systematically implemented diagnostic procedures, incorporating both multicollinearity assessment through variance inflation factor (VIF) examination and parallel line tests, thereby validating the model specification and enhancing the reliability of the regression analysis.

### Data sources

The data used for this analysis originate from the CGSS 2021. It is China's earliest national academic survey project. It has been conducted every year since 2003. In this survey, data are systematically and comprehensively collected at all levels of society in mainland China, the trends of social changes are summarized, and issues of great scientific and practical significance are explored. Moreover, the openness of domestic scientific research and sharing are promoted, data for international comparative research is provided, and this survey serves as a multidisciplinary economic and social data collection platform. The CGSS 2021 data are the latest data currently disclosed by the project team. The data use university research centers, news media, industrial and commercial enterprises, and the NPC to measure public institutional trust.

Therefore, this paper defines the public trust in the NPC as political trust. Research statistics, binary logistic regression, and Ordinal regression are conducted in SPSS22.0.

## Variables

**Dependent variable.** The variables of this paper are the level of political trust. In the CGSS 2021 data, the political trust of Chinese citizens has been assessed by the question "What is your trust in the Chinese People's Congress?" The corresponding code is P54. At the same time, the level of public trust is reflected numerically, with 0 representing "complete distrust" and 10 representing "complete trust". This paper considers missing values that cannot be selected or not selected as unresponsive to political trust questions, while selecting specific trust levels is considered as responding to political trust questions. According to the "Blue Book of Social Mentality: Research Report on Social Mentality in China (2012–2013)," 81–100 points are indicative of "high trust" levels, 71–80 points indicate "basic trust", 61–70 points indicate "trustworthy", 51–60 points indicate "distrust", and 0–50 points indicate "high distrust" [60]. This paper specifically divides the level of political trust into the three levels of distrust, general trust, and high trust. The specific coding is shown in Table 1.

The selection of public trust in the NPC as the primary indicator of political trust in this study is theoretically and institutionally justified. In China's political system, the NPC serves as the highest organ of state power, exercising legislative power and overseeing other key state institutions, including the President, the State Council, the Supreme People's Court, and the Supreme People's Procuratorate. These institutions are not only elected by and accountable to the NPC but also subject to its supervision, making the NPC a central pillar of China's political structure.

The theory of trust transfer posits that trust can be transferred from one entity to another, particularly when an individual's initial trust in an entity is established based on trust in a related entity or within a distinct environment from where the target entity is encountered [61]. The degree of trust transfer is positively correlated with the similarity and interaction between the entities [61]. In the context of China's political system, the NPC, as the supreme state authority, is constitutionally empowered to elect and oversee other state institutions. Given this institutional relationship, public trust in the NPC is likely to influence trust levels in the state institutions it elects and establishes. Conversely, due to the

**Table 1. Variable assignment.**

| Variable | Variable Name | Variable Assignment | Variable Question | Variable Type |
|---|---|---|---|---|
| Dependent Variable | Responsiveness to political trust questions | Be unable to choose = 0, Lack of valuing = 0, 0 = 1, 1 = 1, 2 = 1, 3 = 1, 4 = 1, 5 = 1, 6 = 1, 7 = 1, 8 = 1, 9 = 1, 10 = 1 | P5_4 | Dual Variable |
| | Level of political trust | 0-6 = 1, 7-8 = 2, 9-10 = 3 | | Ordinal Variable |
| Independent Variable | Whether to use the Internet | Never = 0, Seldom = 1, Sometimes = 1, Frequently = 1, Very Frequently = 1 | A28_5 | Dual Variable |
| | Internet use frequency | Never = 0, Several or less a year = 1, Several times a month = 2, Several times a week = 3, Every day = 4 | A30_12 | Ordinal Variable |
| Control Variable | Gender | Female = 0, Male = 1 | A2 | Dual Variable |
| | Age | Investigation year reduction years of birth | A3 | Ratio variable |
| | Political status | Non- CPC member = 0, CPC member = 1 | A10 | Dual Variable |
| | Religious belief | No religious belief = 0, Religious belief = 1 | A5 | Dual Variable |
| | Education level | Never received any education = 0, old-style private school = 0, Primary school = 6, Junior high school = 9, Vocational high school (ordinary high school, technical secondary school, technical school) = 12, Junior college = 15, Bachelor degree = 16, Graduate degree or above = 19 | A7a | Interval variable |
| | Socioeconomic status | The lower layer = 1, middle and lower layers = 2, middle layer = 3, middle and upper layers = 4, upper layer = 5 | A43e | Ordinal Variable |

principal-agent relationship between the NPC and these institutions, public trust in state institutions may also reciprocally affect trust in the NPC. Therefore, it is reasonable for this study to select public trust in the National People's Congress as a measure of political trust for investigation.

**Independent variables.** The independent variables in this paper are "whether to use the Internet" and "Internet use frequency". Internet use refers to chatting, entertainment, shopping, and other activities carried out with various media terminals such as mobile phones, computers, and network televisions with the goal to meet the physical and mental needs of users [62]. The Internet variable is measured in the CGSS 2021 questionnaire by the question "In the past year, can you measure your use of the Internet (including mobile phones)". In the responses to this question, 1 means never, 2 means seldom, 3 means sometimes, 4 means frequently, 5 means very frequently, 98 means "I don't know", 99 means that the respondent refused to answer. This paper eliminates the choices of "unknown" and "refusal to answer" during the analysis process. Regarding Internet use frequency, the item A30_12 of the questionnaire is "In the past year, did you often surf the Internet in your free time" [62]. A response of 5 means never, 4 means several times a year or less, 3 means several times a month, 2 means several times a week, 1 means daily, 98 means that the respondent did not know, and 99 means refusal to answer. This paper eliminates the choice of "unknown" and "refusal to answer" during the analysis process. The specific values are shown in Table 1.

**Control variables.** Considering that the public's political trust is not only affected by macro-level social and economic factors, but also by individual factors such as educational and economic status, this paper finally determines variables such as socioeconomic status as control variables. Education is assigned in reference to existing research [63], and the specific assignment is shown in Table 1.

## Results of the empirical analysis.

### Descriptive statistical analysis

This paper uses SPSS22.0 for a statistical analysis of basic variables (Table 2). In responsiveness to political trust questions, the average value is 0.32 and the median is 0, indicating that many interviewees have not responded to this question. Regarding the level of political trust, only 2,613 respondents reported their trust in the NPC, and their average was 2.62, indicating that the interviewees who provided an answer to this question reported a high degree of trust in the Chinese People's Congress. That is to say, the political trust of the Chinese public is at a high level (the median number is 3), indicating that at least half of respondents maintain a high degree of trust in the NPC. The mean of the variable "Whether to use the Internet" is 0.72, indicating that most interviewees use the Internet. The average value of the Internet use frequency is 2.55, indicating that the Internet use frequency of interviewees is at an average level, and the value ranges between several times a week and several times a month. The median number 4 indicates that at least half of interviewees use the Internet daily.

**Table 2. Variable description statistics.**

| Variable | Average | Median | Standard Deviation | Minimum | Maximum | Valid Value | Missing Value |
|---|---|---|---|---|---|---|---|
| Responsiveness to political trust questions | 0.32 | 0 | 0.47 | 0 | 1 | 8148 | 0 |
| Level of political trust | 2.62 | 3 | 0.66 | 1 | 3 | 2613 | 5535 |
| Whether to use the Internet | 0.72 | 1 | 0.45 | 0 | 1 | 8141 | 7 |
| Internet use frequency | 2.55 | 4 | 1.81 | 0 | 4 | 8127 | 21 |
| Gender | 0.45 | 0 | 0.50 | 0 | 1 | 8148 | 0 |
| Age | 51.64 | 53 | 17.57 | 18 | 99 | 8148 | 0 |
| Political status | 0.12 | 0 | 0.32 | 0 | 1 | 8135 | 13 |
| Religious belief | 0.08 | 0 | 0.26 | 0 | 1 | 8148 | 0 |
| Education level | 9.31 | 9 | 4.73 | 0 | 19 | 8127 | 21 |
| Socioeconomic status | 2.27 | 2 | 0.90 | 1 | 5 | 7969 | 179 |

The average gender value is 0.45, which means that there are many women among respondents, and the proportion of men and women is basically balanced. The average age of respondents is 51.64, and the interviewees are generally relatively older. The median age is 53, indicating that at least half of the interviewees are 53 or older. The average political identity is 0.12, indicating that most of the respondents are not members of the CPC. The average religious belief is 0.08, indicating that religious beliefs are not prevalent among respondents. The average education level is 9.31, and the median is 9, which shows that at least half of the respondents hold junior high school degrees and above, and the maximum value is 19, indicating that some respondents hold a graduate degree and above. The average Socioeconomic status is 2.27, indicating that most interviewees believe that they are in middle and lower income levels. The median number is 2, which further shows that at least half of interviewees believe that their socio-economic status is in the middle and lower levels and below.

### The impact of Internet use on "responsiveness to political trust questions" of the public

Before analyzing the impact of the public's use of the Internet on their responsiveness to political trust questions, first, multiple common linear diagnosis is conducted. The results show that the tolerance exceeded 0.1 and the variance inflation factor (VIF) was less than 5, indicating that there are no serious multiple common linear problems between the variables; then, binary logistic analysis was performed. The significance of Hosemer and Lemeshow in Table 3 is greater than 0.05, indicating that the equation fits better. Table 4 shows that whether to use the Internet is not statistically significant

Table 3. Public Internet use affects the overall test results of "responsiveness to political trust questions" models.

| Model Type | Predictive accuracy(%) | −2LL | Cox & Snell R² | Nagelkerke R ² | Hosemer Lemeshow (sig) |
|---|---|---|---|---|---|
| Model1(N = 7929) | 67.8 | 9946.303 | 0.002 | 0.002 | 0.990 |
| Model2(N = 7916) | 67.8 | 9929.060 | 0.002 | 0.002 | 0.902 |

Table 4. The impact of public Internet use on "responsiveness to political trust questions".

| Variables | Coefficient of regression | | Exp(B) | | 95%EXP(B) Confidence interval | | | |
|---|---|---|---|---|---|---|---|---|
| | | | | | Lower limit | | Upper limit | |
| | Model1 | Model2 | Model1 | Model2 | Model1 | Model2 | Model1 | Model2 |
| Internet use frequency | —— | −0.001 (0.018) | —— | 0.999 | —— | 0.965 | —— | 1.034 |
| Whether to use the Internet | −0.006 (0.068) | —— | 0.994 | —— | 0.87 | —— | 1.136 | —— |
| Gender | 0.068 (0.05) | 0.064 (0.05) | 1.07 | 1.066 | 0.971 | 0.968 | 1.179 | 1.175 |
| Age | −0.003 (0.002) | −0.003 (0.002) | 0.997 | 0.997 | 0.994 | 0.994 | 1.001 | 1.001 |
| Political status | 0.103 (0.079) | 0.099 (0.079) | 1.108 | 1.105 | 0.949 | 0.946 | 1.294 | 1.29 |
| Religious belief | −0.248** (0.097) | −0.244* (0.097) | 0.781 | 0.783 | 0.646 | 0.648 | 0.943 | 0.947 |
| Education level | −0.003 (0.007) | −0.002 (0.007) | 0.997 | 0.998 | 0.984 | 0.985 | 1.011 | 1.011 |
| Socioeconomic status | −0.003 (0.027) | −0.002 (0.027) | 0.997 | 0.998 | 0.945 | 0.946 | 1.052 | 1.053 |
| Constant | −0.606*** (0.157) | −0.61*** (0.156) | 0.546 | 0.543 | | | | |

Note: *p < 0.05

**p < 0.01

*** p < 0.001; Model1 N = 7929 Model 2 N = 7916; the standard error is enclosed in parentheses

(p = 0.933 > 0.05), thus indicating that the use of the Internet does not affect the public's responsiveness to political trust questions; therefore, hypothesis 1 is not verified. At the same time, the significance of the Internet use frequency and the responsiveness to political trust questions is 0.959; therefore, the frequency of Internet use does not affect the responsiveness to political trust questions. Therefore, it is assumed that hypothesis 2 is not verified. Among the control variables, we found that the public's religious belief had a significant negative impact on responsiveness to political trust questions, while age, political status, education level, and socioeconomic status had no significant impact.

### The impact of public internet use on the level of political trust

Multiple common linear analysis reveals that the variables of Model 3 and Model 4 do not have serious multiple common linearity. The ordinal regression model is used to study the impact of Internet use on the level of political trust (Table 5). The parallel line test results of Model 3 and Model 4 in Table 5 are 0.427 and 0.419(p > 0.05). At the same time, the final model is significant at the 0.001 level, indicating that it is better than a model with only intercept. Pearson and deviation are significantly greater than 0.05, indicating that the model fits the data well; therefore, regression analysis is performed (Table 6). In Table 6, the significance level of "whether to use the Internet = 0" in Model 3 is less than 0.001 and the estimated value of "whether to use the Internet = 0" is positive. Therefore, the political trust of the public who do not use the Internet is higher than that of the public who use the Internet, and Hypothesis 3 is verified. In Model 4, the significance of "Internet use frequency = 0" and " Internet use frequency = 3" are 0.000 and 0.010 less than 0.05 respectively. This result means that the level of political trust in respondents who do not use the Internet and those who use it several times a week differs significantly from the level of political trust in people who use the Internet every day. The level of public political trust amog respondents who never use the Internet is much higher than that in people who use the Internet several times a week and every day. Specifically, since the estimated value for Internet use frequency = 0 (0.487) is greater than 0, the political trust of those who never use the internet is higher than that of those who use it daily. The estimated value for Internet use frequency = 3 (−0.403) is less than 0, indicating that the political trust of those who use the internet multiple times a week is lower than that of those who use it daily. There is no significant difference in the political trust of those who use the internet multiple times a year or less, or multiple times a month, compared to those who use it daily. This result shows that the impact of the frequency of public Internet use on political trust is not linear. Among the control variables, there was a significant gender difference in the public's level of political trust in the National People's Congress, with men showing higher levels of trust than women. Other control variables did not have a significant impact on the level of public political trust.

## Discussion and conclusions

### Discussion of findings

By descriptive statistical analysis, binary logistic, and ordinal regression analysis, the research hypothesis verification of the article is shown in Table 7. Specifically, internet use and its frequency do not affect Chinese public's responsiveness to political trust questions regarding the NPC of China, but they do affect the level of public trust in the NPC, albeit in a non-linear manner. There is no significant difference in political trust levels among individuals who use the internet a few

**Table 5. Internet use affects the overall test results of public political trust level models.**

| Model Type | −2LL | chi-square | df | sig | Pearson correlation coefficient | Deviation significance | Cox&Snell significance | Nagelkerke significance | McFadden significance | Parallel Lines significance |
|---|---|---|---|---|---|---|---|---|---|---|
| Model3(N = 2550) | 3334.939 | 70.967 | 10 | 0.000 | 0.405 | 1.000 | 0.027 | 0.035 | 0.018 | 0.427 |
| Model4(N = 2545) | 3468.465 | 71.353 | 13 | 0.000 | 0.534 | 1.000 | 0.028 | 0.035 | 0.018 | 0.419 |

**Table 6. The impact of public Internet use on the level of political trust.**

| Variable | | estimated value | | Wald | | df | | 95%EXP(B) Confidence interval | | | |
|---|---|---|---|---|---|---|---|---|---|---|---|
| | | | | | | | | Lower limit | | Upper limit | |
| | | Model3 | Model4 | Model3 | Model4 | Model3 | Model4 | Model3 | Model4 | Model3 | Model4 |
| **Dependent variable** | [Level of political trust = 1.00] | −2.8* (1.132) | −2.917** (1.134) | 6.119 | 6.622 | 1 | 1 | −5.018 | −5.139 | −0.581 | −0.695 |
| | [Level of political trust = 2.00] | −1.522 (1.131) | −1.642 (1.132) | 1.813 | 2.102 | 1 | 1 | −3.738 | −3.861 | 0.694 | 0.578 |
| **Independent variable** | [Whether to use the Internet =.00] | 0.674*** (0.135) | —— | 25.012 | —— | 1 | —— | 0.41 | —— | 0.938 | —— |
| | [Whether to use the Internet =1.00] | 0a | —— | . | —— | 0 | —— | . | —— | . | —— |
| | [Internet use frequency =.00] | —— | 0.487*** (0.138) | —— | 12.522 | —— | 1 | —— | 0.217 | —— | 0.757 |
| | [Internet use frequency =1.00] | —— | −0.002 (0.282) | —— | 0 | —— | 1 | —— | −0.554 | —— | 0.55 |
| | [Internet use frequency =2.00] | —— | −0.218 (0.249) | —— | 0.765 | —— | 1 | —— | −0.707 | —— | 0.271 |
| | [Internet use frequency =3.00] | —— | −0.403** (0.156) | —— | 6.695 | —— | 1 | —— | −0.708 | —— | −0.098 |
| | [Internet use frequency =4.00] | —— | 0a | —— | . | —— | 0 | —— | . | —— | . |
| **Control variable** | Age | 0.004 (0.003) | 0.005 (0.003) | 1.316 | 2.229 | 1 | 1 | −0.003 | −0.002 | 0.011 | 0.012 |
| | Education Level | −0.003 (0.013) | −0.007 (0.013) | 0.061 | 0.334 | 1 | 1 | −0.028 | −0.033 | 0.022 | 0.018 |
| | [Gender=.00] | −0.243** (0.09) | −0.248** (0.09) | 7.233 | 7.551 | 1 | 1 | −0.42 | −0.425 | −0.066 | −0.071 |
| | [Gender=1.00] | 0a | 0a | . | . | 0 | 0 | . | . | . | . |
| | [Political status=.00] | 0.049 (0.143) | 0.074 (0.143) | 0.116 | 0.268 | 1 | 1 | −0.232 | −0.207 | 0.33 | 0.356 |
| | [Political status=1.00] | 0a | 0a | . | . | 0 | 0 | . | . | . | . |
| | [Religious belief=.00] | −0.021 (0.183) | −0.01 (0.183) | 0.013 | 0.003 | 1 | 1 | −0.38 | −0.369 | 0.338 | 0.349 |
| | [Religious belief=1.00] | 0a | 0a | . | . | 0 | 0 | . | . | . | . |
| | [Socioeconomic status =1.00] | −0.968 (1.074) | −1.049 (1.075) | 0.813 | 0.953 | 1 | 1 | −3.074 | −3.157 | 1.137 | 1.058 |
| | [Socioeconomic status =2.00] | −0.858 (1.075) | −0.969 (1.075) | 0.638 | 0.811 | 1 | 1 | −2.965 | −3.076 | 1.248 | 1.139 |
| | [Socioeconomic status =3.00] | −0.686 (1.074) | −0.788 (1.075) | 0.408 | 0.538 | 1 | 1 | −2.791 | −2.895 | 1.419 | 1.318 |
| | [Socioeconomic status =4.00] | −0.829 (1.087) | −0.942 (1.088) | 0.581 | 0.749 | 1 | 1 | −2.96 | −3.074 | 1.302 | 1.191 |
| | [Socioeconomic status =5.00] | 0a | 0a | . | . | 0 | 0 | . | . | . | . |

Note: *p<0.05

**p<0.01

*** p<0.001; Model 3 N = 2550, Model 4 N = 2545; the standard error is enclosed in parentheses;

Link function: Logit. a: This parameter is zero because it is redundant.

**Table 7. Validation of the research hypotheses.**

| Research hypotheses | Whether the hypothesis is supported |
|---|---|
| **Hypothesis 1:** People who use the Internet are less likely to respond to the question of trust in the NPC of China than people who do not use the Internet. | No |
| **Hypothesis 2:** The higher the frequency of public Internet use, the less likely the public is to respond to the question of trust in the NPC of China. | No |
| **Hypothesis 3:** People who use the Internet have lower levels of political trust in the NPC of China than people who do not use the Internet. | Yes |
| **Hypothesis 4:** The higher the frequency of public Internet use, the lower the political trust of the public in China's NPC. | Partial |

times a year or less, a few times a month, or daily. However, those who never use the internet in their free time exhibit the highest level of political trust, while those who use the internet a few times a week demonstrate the lowest level. At the same time, the study also found that religious belief significantly influence public's responsiveness to political trust questions; those with religious beliefs are even less likely to respond to such questions. Furthermore, there are significant gender differences in the level of political trust among the public, with men exhibiting significantly higher levels of political trust than women.

## Implications for research

The theoretical contributions of this paper are as follows: First, unlike previous studies that solely focused on the impact of internet use on levels of political trust, this paper also considers whether internet use influences how the public's responsiveness to political trust questions. Individuals' political views are classified as sensitive personal information [39,40]. According to privacy calculus, decisions regarding the disclosure of such information involve a risk-benefit analysis [64]. In the context of in-home interviews, a lack of trust between respondents and interviewers consequently led to cautious responses to these questions [65]. Therefore, future theoretical research could investigate the interrelationships among the three core variables: public Internet usage, interpersonal trust levels, and responsiveness to political trust questions.

Second, unlike existing research, it is not the case that more frequent Internet use is associated with lower levels of political trust. One reason for this may be the use of public trust in the NPC as a measure of political trust. The NPC holds an important position in the Chinese political system as one of the most important political institutions. Top political leaders and key political institutions are shielded from negative reports about facts and critical discussions [66]. Therefore, increased time spent online does not necessarily lead to greater exposure to negative information about the NPC. Additionally, discussions in Chinese cyberspace tend to be largely positive, especially concerning central institutions [66]. As a result, more frequent Internet use likely increases exposure to positive news and information. Existing research has also found that the impact of the internet on public political trust is a complex process. Even within internet participation, online information and consultations tend to enhance public political trust, while online decision-making undermines it [29]. Additionally, the influence of internet use on public political trust exhibits significant generational differences, as political trust is shaped by socio-historical contexts [27]. This explains why the frequency of internet use has a nonlinear effect on public political trust: frequency alone cannot capture usage preferences or reflect individuals' personal experiences and social backgrounds. Therefore, future research on internet usage and political trust should move beyond a sole focus on usage frequency to incorporate other relevant factors, such as usage preferences and the values of the respondents' parental generation.

## Implications for practice

First, since the 18th National Congress of the CPC, the CPC and the government have strengthened the management of the Internet, gradually promoted and improved the comprehensive network governance system, and introduced many

laws related to the Internet. The network ecology has fundamentally changed in response. With the Chinese government's strong supports for bridging the Internet access devices and China Mobile Internet technology, more and more individuals have changed their previous role from passive information receivers to creators and publishers of information. The rapid dissemination of public opinion and rumors online makes it easier for people using the Internet to obtain relevant information, which affects their own trust level. However, the rapid development of the Chinese internet has significant advantages., and the Internet has promoted the rapid development of China's digital economy and digital governments. For example, according to the "China Digital Economic Development Research Report (2023)", which was released by the China Institute of Information and Communication, in 2022, the size of China's digital economy reached 5.2 trillion yuan, accounting for 41.5% of China's GDP [67]. The "Digital China Development Report (2022)" released by the China National Internet Information Office showed that the international ranking of the development of China's electronic government index rose from 78 in 2012–43 in 2022. This development identified China as one of the fastest developing countries. The National E-Government Extranet achieves full coverage of prefecture-city and county levels, with a township coverage rate of 96.1% [68]. As a result, Internet technology has promoted the development of China's economy and society, but the Chinese government still needs to take relevant measures (such as theme cultural publicity and improvement of the online ecosystem) to improve the political trust of Internet users.

Second, the public's reluctance to respond questions regarding political trust warrants attention, as this may compromise the accuracy of political trust survey results and the Chinese government's self-assessment. On the one hand, trust affects individuals' willingness to share personal information [35]. When conducting in-home interviews, it is advisable to involve the respondent's relatives or acquaintances whenever possible to foster mutual trust. On the other hand, in China, public trust in local governments carries two dimensions: first, evaluating the commitment and capability of local governments to serve the public interest; second, evaluating the commitment and capability of the central leadership to ensure their local representatives serve the public interest [69]. Research on China has revealed that whenever the public expresses dissatisfaction with local governments, their sentiment toward the national government gradually deteriorates as well. Evaluations of local government issues significantly influence public satisfaction with the central government [70]. Therefore, by surveying and evaluating public trust in local governments, one can also gauge public trust in the central government.

### Limitations and future research

This study has several limitations. First, it focuses exclusively on the public's political trust in the central government of China, without addressing trust in local governments, despite the clear hierarchical trust structure between the Chinese public and central versus local authorities [69]. Second, the study does not incorporate intermediary or moderating variables, such as political values [23], to explore their potential impact on the public's political trust in the context of internet use. Third, this study finds that internet use does not significantly influence public's responsiveness to political trust questions. However, the paper does not delve deeper into this issue for further exploration. Future research should expand its scope to examine political trust not only across central and local governments but also in various national institutions. Additionally, it should integrate relevant moderating and mediating variables and employ advanced analytical methods, such as structural equation modeling, to conduct more in-depth investigations.

### Supporting information

**S1 File. Annex to the questionnaire.**
(PDF)

**S2 File. Data.**
(XLSX)

## Author contributions

**Conceptualization:** Yanan Wen, Lu Fang.

**Formal analysis:** Lu Fang.

**Funding acquisition:** Yanan Wen, Xue Sun.

**Methodology:** Xue Sun, Lu Fang.

**Writing – original draft:** Yanan Wen, Xue Sun, Lu Fang.

**Writing – review & editing:** Xue Sun, Lu Fang.

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
