## [Decision Letter · Decision Letter 0]

16 May 2025

Dear Dr. FANG,

Thank you for submitting your manuscript to PLOS ONE. After careful consideration, we feel that it has merit but does not fully meet PLOS ONE’s publication criteria as it currently stands. Therefore, we invite you to submit a revised version of the manuscript that addresses the points raised during the review process.

We look forward to receiving your revised manuscript.

Kind regards,

Floris Vermeulen

Academic Editor

PLOS ONE

**Journal Requirements:**

1. When submitting your revision, we need you to address these additional requirements. Please ensure that your manuscript meets PLOS ONE's style requirements, including those for file naming. The PLOS ONE style templates can be found at https://journals.plos.org/plosone/s/file?id=wjVg/PLOSOne_formatting_sample_main_body.pdf and https://journals.plos.org/plosone/s/file?id=ba62/PLOSOne_formatting_sample_title_authors_affiliations.pdf 2. Please note that PLOS ONE has specific guidelines on code sharing for submissions in which author-generated code underpins the findings in the manuscript. In these cases, we expect all author-generated code to be made available without restrictions upon publication of the work. Please review our guidelines at https://journals.plos.org/plosone/s/materials-and-software-sharing#loc-sharing-code and ensure that your code is shared in a way that follows best practice and facilitates reproducibility and reuse. 3. Thank you for uploading your study's underlying data set. Unfortunately, the repository you have noted in your Data Availability statement does not qualify as an acceptable data repository according to PLOS's standards. At this time, please upload the minimal data set necessary to replicate your study's findings to a stable, public repository (such as figshare or Dryad) and provide us with the relevant URLs, DOIs, or accession numbers that may be used to access these data. For a list of recommended repositories and additional information on PLOS standards for data deposition, please see https://journals.plos.org/plosone/s/recommended-repositories. 4. We note that this data set consists of interview transcripts. Can you please confirm that all participants gave consent for interview transcript to be published? If they DID provide consent for these transcripts to be published, please also confirm that the transcripts do not contain any potentially identifying information (or let us know if the participants consented to having their personal details published and made publicly available). We consider the following details to be identifying information:

- Names, nicknames, and initials

- Age more specific than round numbers

- GPS coordinates, physical addresses, IP addresses, email addresses

- Information in small sample sizes (e.g. 40 students from X class in X year at X university)

- Specific dates (e.g. visit dates, interview dates)

- ID numbers Or, if the participants DID NOT provide consent for these transcripts to be published:

- Provide a de-identified version of the data or excerpts of interview responses

- Provide information regarding how these transcripts can be accessed by researchers who meet the criteria for access to confidential data, including:

a) the grounds for restriction

b) the name of the ethics committee, Institutional Review Board, or third-party organization that is imposing sharing restrictions on the data

c) a non-author, institutional point of contact that is able to field data access queries, in the interest of maintaining long-term data accessibility.

d) Any relevant data set names, URLs, DOIs, etc. that an independent researcher would need in order to request your minimal data set. For further information on sharing data that contains sensitive participant information, please see: https://journals.plos.org/plosone/s/data-availability#loc-human-research-participant-data-and-other-sensitive-data If there are ethical, legal, or third-party restrictions upon your dataset, you must provide all of the following details (https://journals.plos.org/plosone/s/data-availability#loc-acceptable-data-access-restrictions):

a) A complete description of the dataset

b) The nature of the restrictions upon the data (ethical, legal, or owned by a third party) and the reasoning behind them

c) The full name of the body imposing the restrictions upon your dataset (ethics committee, institution, data access committee, etc)

d) If the data are owned by a third party, confirmation of whether the authors received any special privileges in accessing the data that other researchers would not have

e) Direct, non-author contact information (preferably email) for the body imposing the restrictions upon the data, to which data access requests can be sent 5. Please include captions for your Supporting Information files at the end of your manuscript, and update any in-text citations to match accordingly. Please see our Supporting Information guidelines for more information: http://journals.plos.org/plosone/s/supporting-information.

**Additional Editor Comments:**

The reviewer sees merit in your manuscript, but also identifies some major issues that needs to be resolved before publication in PLOS can be considered again.

Reviewers' comments:

Reviewer's Responses to Questions

**Comments to the Author**

1. Is the manuscript technically sound, and do the data support the conclusions?

Reviewer #1: Partly

2. Has the statistical analysis been performed appropriately and rigorously?

Reviewer #1: Yes

3. Have the authors made all data underlying the findings in their manuscript fully available?

Reviewer #1: Yes

4. Is the manuscript presented in an intelligible fashion and written in standard English?

Reviewer #1: Yes

**Reviewer #1:** Thank you for the opportunity to review the article entitled “The impact of internet use in the digital era on public political trust in China”. The article draws on data from China’s General Social Survey (CGSS) 2021 to explore how political trust in China is impacted by Internet use. The two main findings of the authors are that 1) a substantial share of the respondents do not answer the survey question on political trust, and 2) that (frequent) use of the Internet increases the likelihood that lower levels of political trust are found.

Not all parts of the theoretical section, hypotheses, data and conclusions align. The relationship between internet usage and political trust is discussed in the theoretical section, connected to hypotheses 2 and 4, and discussed in the conclusion section. However, the part on why respondents have not answered the survey question on political trust is not grounded in theory. This makes that hypothesis 1 and 3 seem to come out of nowhere and leave the reader to wonder why not answering the survey question about political trust would be linked to internet usage. In addition, the authors draw some conclusions that are not supported by the data that they present:

- Line 346: “These results show that as the public uses the Internet, more unofficial information and unverified information are obtained, which leads to lower level of political trust”. The authors do not measure the content of the Internet usage, so based on the data no conclusions can be drawn about the information that respondents receive online.

- Line 352: “The level of public political trust in respondents who never use the Internet is much higher than that in people who use the Internet several times a week and every day”. As the refence (baseline) category is ‘every day’ you can only say something about ‘every day’ vs. ‘several times a week’ and ‘every day’ vs. ‘never’. You cannot compare ‘never’ vs. ‘several times a week’ as neither of these categories is the reference category.

- Line 374: “This may be attributed to the public’s heightened focus on economic development, driven by the growth of China’s socialist market economy”. There are no sources or data presented that support this claim. If making such a statement, it would be good to reference to sources.

- Line 396: “Specifically, exposure to news on the internet significantly promotes political participation, while recreational use of the internet negatively affects political participation. Thus, this paper further enriches the understanding of how internet use influences political apathy.” The paper does not measure content of internet usage and also does not measure the concept of political apathy, so these conclusions are not supported by the data.

- Line 415: “Third, this study also validates the perspective that even with the establishment of a well-developed gatekeeping system, prolonged exposure to positive information on the internet can still reduce political trust”. Again, the paper does not measure content of internet usage, so this conclusion cannot be derived from the data.

- Line 449: “It is necessary to severely punish illegal acts that may affect public political participation, and to ensure the public’s successful performance of their own political rights.” This is quite a strong statement, how are ‘illegal acts’ defined and what is understood by ‘severely punish’?

The authors have conducted descriptive statistics, binary logistic regression analyses, and ordinal regression analyses. Table 1 is really helpful and provides clear information on the variables that are included in the analyses. Some comments:

- The terminology ‘political trust choice’ is a bit confusing. It would be good to rename this variable to something that makes it clear what is measured. For instance, ‘answer political trust’.

- The authors could provide more information on the CGSS. For instance, how is the survey conducted (telephone, digital, by mail, in person etc.)? Is participation anonymous? This may impact why respondents do or do not answer the question about political trust.

- Effect sizes can be presented to illustrate the impact of the significant results that are found

- The layout of the tables (especially 4 and 6) should be changed, as the tables are currently difficult to read. For instance, by presenting one single column of variables on the left, adding model x and model y as columns and presenting the coefficients and standard errors for each variable in each model. The significance levels can be presented as stars (* p<0.05 ** p<0.01 *** p<0.001)

- It looks like the dependent variables in Table 4 are continuous, while the same dependent variables in Table 6 are categorical. Why is that?

- Table 6 discusses model3 and model4, shouldn’t this be model5 and model6? As model3 and model4 are already discussed in Table 4.

Some final general remarks:

- Line 37: “The Chinese People’s Congress system is an effective form of political organization that is in line with China’s national conditions”. What does this mean?

- Line 200: “Clicks, likes, and shares lead to personalized recommendations that challenge the beliefs of users.” Is it challenge or affirm the beliefs of users?

- Line 89: typo, two periods/full stops

**Do you want your identity to be public for this peer review?** For information about this choice, including consent withdrawal, please see our Privacy Policy

Reviewer #1: No

---

## [Author Response · Author response to Decision Letter 1]

3 Aug 2025

Manuscript ID: PONE-D-25-01950

Title: The Impact of Internet Use in the Digital Era on Public Political Trust in China—An Empirical Study Based on CGSS 2021 Data

Response to Reviewer #1 Comments

We thank Reviewer #1 for their careful reading and constructive suggestions, which have greatly helped us improve the manuscript. Below are our point-by-point responses. All modifications are made using revision mode.

Comment 1 (Reviewer #1): “Not all parts of the theoretical section, hypotheses, data and conclusions align. The relationship between internet usage and political trust is discussed in the theoretical section, connected to hypotheses 2 and 4, and discussed in the conclusion section. However, the part on why respondents have not answered the survey question on political trust is not grounded in theory. This makes that hypothesis 1 and 3 seem to come out of nowhere and leave the reader to wonder why not answering the survey question about political trust would be linked to internet usage.”

Response: We sincerely agree. We have added relevant discussion in lines 197 to 207 to make the proposal of hypothesis 1 more reasonable. Here, we introduce the concept of political participation, identify the factors influencing political participation, and explain that the CGSS uses in-home interviews format, which can hinder responses to these questions due to low levels of trust between respondents and interviewers. We also cite relevant literature demonstrating that the development of the internet has heightened awareness of personal privacy and that social media users are more concerned with protecting their online information than non-users. Furthermore, in lines 240-246, we cite relevant research indicating that frequent internet use negatively impacts social trust, which is correlated with political trust. Furthermore, due to the CGSS survey format, internet users are reluctant to participate in political questions. Furthermore, based on the argument that algorithmic technology and internet use exacerbate the decline of traditional political participation, we propose Hypothesis 3.

Comment 2 (Reviewer #1): “These results show that as the public uses the Internet, more unofficial information and unverified information are obtained, which leads to lower level of political trust”. The authors do not measure the content of the Internet usage, so based on the data no conclusions can be drawn about the information that respondents receive online.

Response: This suggestion is crucial for us to improve our paper. In lines 380 to 384, we modified our statement so that we only base our argument on the results of empirical analysis, arguing that the public who use the Internet has lower political trust, and Hypothesis 2 is established.

Comment 3 (Reviewer #1): “The level of public political trust in respondents who never use the Internet is much higher than that in people who use the Internet several times a week and every day”. As the refence (baseline) category is ‘every day’ you can only say something about ‘every day’ vs. ‘several times a week’ and ‘every day’ vs. ‘never’. You cannot compare ‘never’ vs. ‘several times a week’ as neither of these categories is the reference category.

Response: We fully agree with this amendment. We only compare other categories with ‘every day’ to observe the significance and coefficients, and then determine the impact of Internet use frequency on political trust.

Comment 4 (Reviewer #1): “This may be attributed to the public’s heightened focus on economic development, driven by the growth of China’s socialist market economy”. There are no sources or data presented that support this claim. If making such a statement, it would be good to reference to sources.

Response: This comment helped us discover our omissions. We first added a reference to support our view in line 414. We also added a note from line 414 to line 415 about how the CGSS survey format might affect the public's response rate.

Comment 5 (Reviewer #1): “Specifically, exposure to news on the internet significantly promotes political participation, while recreational use of the internet negatively affects political participation. Thus, this paper further enriches the understanding of how internet use influences political apathy.” The paper does not measure content of internet usage and also does not measure the concept of political apathy, so these conclusions are not supported by the data.

Response: We support this amendment. We have made modifications from lines 436 to 438. After introducing the concept of political apathy, we directly point out that this study found that whether or not to use the Internet and the frequency of Internet use do not affect political apathy. This will help future researchers focus on the relationship between other areas such as Internet usage preferences and political apathy.

Comment 6 (Reviewer #1): “Third, this study also validates the perspective that even with the establishment of a well-developed gatekeeping system, prolonged exposure to positive information on the internet can still reduce political trust”. Again, the paper does not measure content of internet usage, so this conclusion cannot be derived from the data.

Response: After carefully considering the experts' suggestions, we found that our article results could not lead to such a conclusion. In order to ensure the scientificity and rigor of the research paper, we directly deleted this statement in line 545.

Comment 7 (Reviewer #1): “It is necessary to severely punish illegal acts that may affect public political participation, and to ensure the public’s successful performance of their own political rights.” This is quite a strong statement, how are ‘illegal acts’ defined and what is understood by ‘severely punish’?

Response: This comment does point out our problem, but it is difficult to clearly define its scope within the limited space. Therefore, we have directly deleted this statement in line 485 and directly proposed some feasible suggestions.

Comment 8 (Reviewer #1): “The terminology ‘political trust choice’ is a bit confusing. It would be good to rename this variable to something that makes it clear what is measured. For instance, ‘answer political trust’.”

Response: We fully agree with the expert’s opinion and we have changed ‘political trust choice’ to ‘answer political trust’.

Comment 9 (Reviewer #1): The authors could provide more information on the CGSS. For instance, how is the survey conducted (telephone, digital, by mail, in person etc.)? Is participation anonymous? This may impact why respondents do or do not answer the question about political trust.

Response: The author retrieved relevant news reports on CGSS2021 and determined the data obtained through home visits adopted by CGSS2021. We have also provided the source of the relevant news reports in the form of footnotes on page 6, which will help readers have a deeper understanding of this project.

Comment 10 (Reviewer #1): “Effect sizes can be presented to illustrate the impact of the significant results that are found.”

Response: This is an important observation. We report the regression coefficients, exp(B), and other values in the binary logistic analysis to make the results more visible. At the same time, we report indicators such as estimated values in the ordinal regression analysis.

Comment 11 (Reviewer #1): “The layout of the tables (especially 4 and 6) should be changed, as the tables are currently difficult to read. For instance, by presenting one single column of variables on the left, adding model x and model y as columns and presenting the coefficients and standard errors for each variable in each model. The significance levels can be presented as stars (* p<0.05 ** p<0.01 *** p<0.001).”

Response: This comment was crucial in improving our table layout. We adjusted Tables 4 and 6 accordingly, placing the variables on the left and Model x and Model y in the columns. Standard errors were placed within the regression coefficients or estimates, presented separately in parentheses. Significance was also indicated with “*”. This made the tables more visually appealing and intuitive, helping readers better understand the empirical results.

Comment 12 (Reviewer #1): “It looks like the dependent variables in Table 4 are continuous, while the same dependent variables in Table 6 are categorical. Why is that?”

Response: Dear experts! Table 4 presents the impact of internet use on "answer political trust." "Answer political trust" is a dichotomous variable, categorized as 0 or 1, and therefore represents the results of a binary logistic analysis. Table 6, on the other hand, presents the impact of internet use on the level of political trust, which is categorized as 1, 2, and 3. It also presents the results of an ordinal regression analysis.

Comment 13 (Reviewer #1): “It looks like the dependent variables in Table 4 are continuous, while the same dependent variables in Table 6 are categorical. Why is that?”

Response: Dear experts! Table 3 presents the prerequisite results for whether a binary logistic analysis can be conducted. If the relevant indicators in Table 3 are met, a binary logistic analysis can be conducted. We initially intended to combine the results of Tables 3 and 4, but considering that the combined results would be too large, we have to present the results in two tables. Therefore, Tables 3 and 4 present the results of Model 1 and Model 2. Table 5 reports the premises for conducting an ordinal regression analysis, and Table 6 reports the results of an ordinal regression analysis of the impact of internet use on political trust. Both Tables 5 and 6 present the results of Models 3 and 4.

Comment 14 (Reviewer #1): “The Chinese People’s Congress system is an effective form of political organization that is in line with China’s national conditions”. What does this mean?

Response: We have modified the sentence from lines 47 to 49, and we describe it as a political system arrangement that conforms to China's national conditions.

Comment 15 (Reviewer #1): “Clicks, likes, and shares lead to personalized recommendations that challenge the beliefs of users.” Is it challenge or affirm the beliefs of users?

Response: We would like to thank the expert for pointing out this error. After checking the references, we found that it does indeed affirm the user's idea. We have made changes in line 225.

Comment 16 (Reviewer #1): “typo, two periods/full stops.”

Response: We would like to thank the expert for pointing out our spelling error. We have deleted the redundant symbols in line 101 and checked the rest of the text.

Finally, we would like to thank the experts again for their efforts in improving this paper and enhancing its quality.

---

## [Decision Letter · Decision Letter 1]

11 Sep 2025

Dear Dr. Fang,

We look forward to receiving your revised manuscript.

Kind regards,

Floris Vermeulen

Academic Editor

PLOS ONE

Journal Requirements:

Reviewers' comments:

Reviewer's Responses to Questions

**Comments to the Author**

Reviewer #1: (No Response)

2. Is the manuscript technically sound, and do the data support the conclusions?

Reviewer #1: Partly

3. Has the statistical analysis been performed appropriately and rigorously?

Reviewer #1: Yes

4. Have the authors made all data underlying the findings in their manuscript fully available?

Reviewer #1: Yes

5. Is the manuscript presented in an intelligible fashion and written in standard English?

Reviewer #1: Yes

Reviewer #1: Thank you for the opportunity to review the revised version of “The impact of internet use in the digital era on public political trust in China”. The authors have addressed several of the concerns that were raised in the first round of review. I commend the authors for the work that they did. However, I believe there still are some issues that make that at this stage the paper is not ready for publication.

An important issue is that there is a disconnect between the argument on political apathy and the results and conclusions that are presented. The concept of political apathy plays a central role in the conclusions presented by the authors, but from the theoretical and methods section it is not clear how political apathy is conceptualized. The authors argue that political trust manifests at a deeper level as political apathy, and that therefore not answering a survey question about political trust is an indicator of political apathy. This argument requires clarification and elaboration to convince the reader why this is an indicator of political apathy.

The authors have made adjustments to clarify hypotheses one and three, but the link between internet usage and not answering a question about political trust remains unclear. The concept of political participation is introduced, and it is stated that political participation has changed due to internet usage. However, the authors then argue that people may be less likely to answer questions about political trust due to the in-person interviews (as they would be more likely to want to protect their privacy due to internet usage). But the question that emerges is: do people not answer the question due to their internet usage or due to the way in which the data was collected (in-person interviews)?

It is very informative that the authors have added information about the sample collection methods of the CGSS2021 survey. However, the fact that the survey was conducted by means of in-person interviews at home, might be a quite impactful variable (as the authors mention that questions about political trust can be regarded as sensitive). Based on the argumentation the link between internet usage and not answering questions about political trust remains vague. I believe the questions should be: are people more likely to protect their privacy due to internet usage? If they do, it can be expected that they are more likely not to be willing to answer questions about political trust during in-person interviews.

In sum, the authors have to make explicit why not answering questions about political trust is an indicator of political apathy. In addition, the authors have to provide a convincing case on how internet usage affects not answering political trust questions and how they measure this impact.

**Do you want your identity to be public for this peer review?** For information about this choice, including consent withdrawal, please see our Privacy Policy

Reviewer #1: No

---

## [Author Response · Author response to Decision Letter 2]

26 Oct 2025

Rebuttal Letter for Manuscript ID: PONE-D-25-01950R1

Part 1: Cover Letter to the Editor

Lu Fang

University of Electronic Science and Technology of China

Email: fnglu1017@sina.com

Revised Manuscript Submission (Manuscript ID: PONE-D-25-01950R1) - Response to Reviewers' Comments

Dear editor,

Thank you for giving us the opportunity to revise your manuscript entitled “The Impact of Internet Use in the Digital Era on Public Political Trust in China—An Empirical Study Based on CGSS 2021 Data” (Manuscript ID: PONE-D-25-01950R1]. We sincerely appreciate the time and effort invested by you and the reviewers in evaluating our work. We found the reviewers' comments to be constructive, insightful, and invaluable for improving the quality and impact of our paper.

We have carefully considered all points raised by the reviewers and have made extensive revisions to the manuscript accordingly. A detailed, point-by-point response to each specific comment from each reviewer is provided in the attached document. The main categories of revisions we have undertaken include:

Based on the actual situation of this survey and considering that it is inappropriate to use the term "political apathy" in the article, we deleted the expression of political indifference in the article. Instead, based on the existing literature that Internet use affects interpersonal trust and trust affects the willingness to share personal information, we re-organized the "research hypothesis" part and the "discussion and conclusion" part of the article.

Comprehensive Reference Updates: References have been checked, updated where necessary, and formatted according to journal guidelines. Our commentary on cited works is now more substantial and critical.

All changes in the revised manuscript were tracked using the Track Changes feature in Microsoft Word. We firmly believe that these revisions significantly improved the manuscript's quality in terms of language, clarity, scientific depth, logical structure, presentation of figures and tables, and accuracy of references, effectively addressing the core concerns raised by the reviewers.

We respectfully request that you reconsider our revised manuscript for publication in PLOS ONE.

We are fully committed to further refining this work and are more than willing to undertake any additional revisions deemed necessary to meet the high standards of this esteemed journal.

Thank you again for your time and consideration. We look forward to your further guidance.

Sincerely,

Lu Fang

University of Electronic Science and Technology of China

Email: fnglu1017@sina.com

Part 2: Detailed Point-by-Point Response to Reviewers' Comments

Manuscript ID: PONE-D-25-01950R1

Title: The Impact of Internet Use in the Digital Era on Public Political Trust in China—An Empirical Study Based on CGSS 2021 Data

Response to Reviewer #1 Comments

We thank Reviewer #1 for their careful reading and constructive suggestions, which have greatly helped us improve the manuscript. Below are our point-by-point responses. All modifications are made using revision mode.

Comment 1 (Reviewer #1): “Thank you for the opportunity to review the revised version of “The impact of internet use in the digital era on public political trust in China”. The authors have addressed several of the concerns that were raised in the first round of review. I commend the authors for the work that they did. However, I believe there still are some issues that make that at this stage the paper is not ready for publication.

An important issue is that there is a disconnect between the argument on political apathy and the results and conclusions that are presented. The concept of political apathy plays a central role in the conclusions presented by the authors, but from the theoretical and methods section it is not clear how political apathy is conceptualized. The authors argue that political trust manifests at a deeper level as political apathy, and that therefore not answering a survey question about political trust is an indicator of political apathy. This argument requires clarification and elaboration to convince the reader why this is an indicator of political apathy.

The authors have made adjustments to clarify hypotheses one and three, but the link between internet usage and not answering a question about political trust remains unclear. The concept of political participation is introduced, and it is stated that political participation has changed due to internet usage. However, the authors then argue that people may be less likely to answer questions about political trust due to the in-person interviews (as they would be more likely to want to protect their privacy due to internet usage). But the question that emerges is: do people not answer the question due to their internet usage or due to the way in which the data was collected (in-person interviews)?

It is very informative that the authors have added information about the sample collection methods of the CGSS2021 survey. However, the fact that the survey was conducted by means of in-person interviews at home, might be a quite impactful variable (as the authors mention that questions about political trust can be regarded as sensitive). Based on the argumentation the link between internet usage and not answering questions about political trust remains vague. I believe the questions should be: are people more likely to protect their privacy due to internet usage? If they do, it can be expected that they are more likely not to be willing to answer questions about political trust during in-person interviews.

In sum, the authors have to make explicit why not answering questions about political trust is an indicator of political apathy. In addition, the authors have to provide a convincing case on how internet usage affects not answering political trust questions and how they measure this impact.”

Response: We are very grateful for pointing out the problems with our article and we fully agree with it. Based on your comments, combined with a review of relevant literature and the CGSS survey methodology, we have abandoned our previous interpretation of not answering the political trust question as political apathy and answering it as political participation. This is inappropriate given the current data and the definitions of political apathy and political participation.

Your suggestion to modify the article from a privacy perspective provides us with a good idea. First, in the research hypothesis section, we combined Hypotheses 1 and 3, which originally addressed answering the political trust question, and renumbered Hypothesis 3 as Hypothesis 2. We also added the following before Hypotheses 1 and 2:

“According to relevant CGSS reports [For related news reports, see "Summer Semester 100 Village Survey: 2021 Chinese General Social Survey (CGSS)" at: https://spm.ncu.edu.cn/tsgz/bcdc/1c11c5be93544dfc9cc3935d8ab9d635.htm;

"The China Survey and Data Center 2021 Report Card is released!" The website is: https://mp.weixin.qq.com/s?__biz=MzU3NTAwNjk4MA==&mid=2247493929&idx=1&sn=58a9b87ec3dee244e37a0f468e7b37a5&chksm=fd2b1942ca5c9054b7e853ad6226df4b3325b72ee8e763c0581f6960dd26d2a2fdb39248d4e7&scene=27.], the 2021 CGSS employed in-home interviews for data collection. This raises the issue of the respondents' trust in the interviewers [For related news reports, see "Summer Semester 100 Village Survey: 2021 Chinese General Social Survey (CGSS)" at: https://spm.ncu.edu.cn/tsgz/bcdc/1c11c5be93544dfc9cc3935d8ab9d635.htm.]. Interpersonal trust encompasses multiple dimensions, ranging from trust in relatives, friends, and acquaintances to trust in strangers[32]. Interpersonal trust entails positive expectations regarding the behavior of others, thereby motivating individuals to engage in interactions through sharing, mutual assistance, and cooperation[33], while also enhancing their willingness to disclose personal information[34]. Trust is essentially the willingness to take the risk of information disclosure[35]. Political trust refers to the public's subjective evaluation and attitude toward political institutions and politicians[36]. An article on the official website of the Cyberspace Administration of China classifies political opinions and other personal information as sensitive[37]. Similarly, the European Union's General Data Protection Regulation (GDPR) specifies that individuals' political opinions constitute special categories of personal data[38]. Thus, from this vantage point, individuals' responses to questions regarding political trust fall within the scope of personal privacy.

However, Chinese scholars have utilized data from the CGSS 2013 and CGSS 2015 surveys to demonstrate that Internet use exerts a significant negative effect on public trust in strangers[39]. Internet use diminishes respondents' trust in interviewers, thereby reducing the propensity for sharing, mutual assistance, and cooperation between them. Trust fundamentally entails the willingness to assume the risks associated with information disclosure[35]; consequently, owing to the absence of trust between them, respondents are reluctant to respond to questions pertaining to political trust. Drawing on the foregoing analysis, we advance the following two hypotheses.”

Now, I'd like to share with you my modification ideas. After reviewing relevant laws and regulations, we found that personal political views are indeed sensitive personal information. Therefore, answering questions about political trust involves the disclosure of personal privacy. A review of the literature on personal privacy disclosure reveals that trust and perceived risk are important factors. Existing research has shown that internet use affects interpersonal trust, with internet use significantly negatively impacting trust in strangers. Furthermore, trust is essentially an individual's willingness to assume the risk of disclosing information. News reports on the CGSS survey suggest a lack of trust between respondents and interviewers, leading to a potential reluctance to assume the risk of disclosing information. Based on this, we propose Hypotheses 1 and 2.

The “implications for research” section in the “discussion and conclusions” of the article, we also deleted the previous content on political apathy and political participation. Instead, we discussed Internet use, interpersonal trust, and private information disclosure.

In the “Implications for practice” section, we rewrote the second paragraph, which discusses research methods and approaches to ensure that the public is more willing to answer questions about political trust.

Finally, we thank you again for your efforts in improving this article and enhancing its quality.

---

## [Decision Letter · Decision Letter 2]

18 Nov 2025

Dear Dr. Fang,

Thank you for submitting your manuscript to PLOS ONE. After careful consideration, we feel following the minor comments by the reviewer that it has merit but does not fully meet PLOS ONE’s publication criteria as it currently stands. Therefore, we invite you to submit a revised version of the manuscript that addresses the points raised during the review process.

We look forward to receiving your revised manuscript.

Kind regards,

Floris Vermeulen

Academic Editor

PLOS ONE

Journal Requirements:

Reviewers' comments:

Reviewer's Responses to Questions

**Comments to the Author**

Reviewer #1: (No Response)

2. Is the manuscript technically sound, and do the data support the conclusions?

Reviewer #1: Partly

3. Has the statistical analysis been performed appropriately and rigorously?

Reviewer #1: Yes

4. Have the authors made all data underlying the findings in their manuscript fully available?

Reviewer #1: Yes

5. Is the manuscript presented in an intelligible fashion and written in standard English?

Reviewer #1: Yes

Reviewer #1: Thank you for the opportunity to review the revised version of “The Impact of Internet Use in the Digital Era on Public Political Trust in China – An Empirical Study Based on CGSS 2021 Data”. The authors made revisions in line with the suggestions (expand on link between internet usage and political trust, revise usage of the concept political apathy, and elaborate on impact of sample collection methods), and the manuscript has improved as a result. Still some (minor) observations remain:

- Throughout the paper the authors deploy the terms “choice of political trust” and “answer political trust”. These terms are confusing as it is not clear what these concepts entail. The authors should either replace these terms (for instance: responsiveness to political trust questions) or define “choice of political trust”/”answer political trust” from the start.

- Line 111-115: this section discusses the impact of factors on political trust. One of these factors is described as ‘income’ (“… income reduces trust in mature democracies”). Shouldn’t this be income level? And is it then that lower or higher income levels reduce political trust?

- Line 143-144: “In the new information environment created by the Internet, political trust has been strengthened…” Has should be can

- Line 346: “average education level is 9.31, and the average is 9,…” Should be “and the median is 9”

- Line 380-384: “level of political trust in people/respondents” should be “level of political trust among people/respondents”

- In the discussion section some additional findings are discussed. Additional finding 2 and 3 do not seem to be derived from the data analyses, but rather seem to be conclusions based on literature research. If this is not the case and these numbers are actually derived from the data analyses, the authors should discuss these findings in the data section.

**Do you want your identity to be public for this peer review?** For information about this choice, including consent withdrawal, please see our Privacy Policy

Reviewer #1: No

---

## [Author Response · Author response to Decision Letter 3]

1 Dec 2025

Detailed Point-by-Point Response to Reviewers' Comments

Manuscript ID: PONE-D-25-01950R2

Title: The Impact of Internet Use in the Digital Era on Public Political Trust in China—An Empirical Study Based on CGSS 2021 Data

Response to Reviewer #1 Comments

We thank Reviewer #1 for their careful reading and constructive suggestions, which have greatly helped us improve the manuscript. Below are our point-by-point responses. All modifications are made using revision mode.

Comment 1 (Reviewer #1): Throughout the paper the authors deploy the terms "choice of political trust" and "answer political trust". These terms are confusing as it is not clear what these concepts entail. The authors should either replace these terms (for instance: responsiveness to political trust questions) or define "choice of political trust"/"answer political trust" from the start.

Response 1: We highly appreciate the reviewer's comment, as it is of great help in improving our paper and in assisting readers. We adopted the reviewers' suggestions to replace the terms "choice of political trust" and "answer political trust" with "responsiveness to political trust questions". We added "This necessitates that this article pay attention to questions about public responsiveness to political trust. Therefore, this paper examines the impact of internet use on public political trust, encompassing both its influence on responsiveness to political trust questions and its effect on the overall level of public political trust." after "At the same time, an interesting phenomenon was also discovered, namely that among the total sample of 8,148 respondents, only 2,613 responded to relevant questions, while nearly 2/3 of all respondents did not respond this question." In order to help readers better understand responsiveness to political trust.

Comment 2 (Reviewer #1): Line 111-115: this section discusses the impact of factors on political trust. One of these factors is described as ‘income’ ("… income reduces trust in mature democracies"). Shouldn’t this be income level? And is it then that lower or higher income levels reduce political trust?

Response 2: We appreciate your meticulous pointing out of this issue. We reread the paper containing this viewpoint, and indeed, income here refers to income level, and in mature democracies, high income levels can actually lead to low political trust. At the same time, we have made corresponding modifications to the original text, changing it to "on the contrary, post-materialism, political radicalism, corruption indulgence, and income level impede political trust[21]. In other words, in mature democracies, the higher an individual's income level, the lower their level of political trust. At the same time, income level has different effects in different countries. While income level reduces trust in mature democracies, it increases trust in countries of Latin America and Eastern Europe, but has no effect on the countries of the former Soviet Union[21]."

Comment 3 (Reviewer #1): Line 143-144: "In the new information environment created by the Internet, political trust has been strengthened…" Has should be can

Response 3: Thank you for your suggestion. We have revised the original text from "In the new information environment created by the Internet, political trust has been strengthened" to "In the new information environment created by the Internet, political trust can been strengthened’

Comment 4 (Reviewer #1): Line 346: "average education level is 9.31, and the average is 9,…" Should be "and the median is 9"

Response 4: Thank you for pointing out this error. We located the original text and changed the average to the median.

Comment 5 (Reviewer #1): Line 380-384: "level of political trust in people/respondents" should be "level of political trust among people/respondents"

Response 5: Thank you so much for pointing out this error. We have corrected that part of the original text.

Comment 6 (Reviewer #1): In the discussion section some additional findings are discussed. Additional finding 2 and 3 do not seem to be derived from the data analyses, but rather seem to be conclusions based on literature research. If this is not the case and these numbers are actually derived from the data analyses, the authors should discuss these findings in the data section.

Response 6: Thank you for your suggestion. Indeed, we should not draw upon viewpoints from other literature here. We have made the following adjustments.

First, we removed the four additional findings from the original proposal and retained only the findings obtained from the article's data analysis. We changed the value after "...while those who use the internet a few times a week demonstrate the lowest level" to "At the same time, the study also found that religious belief significantly influence public's responsiveness to political trust questions; those with religious beliefs are even less likely to respond to such questions. Furthermore, there are significant gender differences in the level of political trust among the public, with men exhibiting significantly higher levels of political trust than women".

Second, in the section "The impact of Internet use on "responsiveness to political trust questions" of the public". We added "Among the control variables, we found that the public's religious belief had a significant negative impact on responsiveness to political trust questions, while age, political status, education level, and socioeconomic status had no significant impact" after "it is assumed that hypothesis 2 is not verified".

Third, in the section "The impact of public Internet use on the level of political trust". We added "Among the control variables, there was a significant gender difference in the public's level of political trust in the National People's Congress, with men showing higher levels of trust than women. Other control variables did not have a significant impact on the level of public political trust" after "This result shows that the impact of the frequency of public Internet use on political trust is not linear".

Finally, we read the entire text again and, without changing the existing structure and content, we adjusted some sentences to ensure there were no grammatical or spelling errors.

We thank you again for your efforts in improving this article and enhancing its quality.

Kind regards.

---

## [Editor Report · Decision Letter 3]

15 Dec 2025

The Impact of Internet Use in the Digital Era on Public Political Trust in China—An Empirical Study Based on CGSS 2021 Data

PONE-D-25-01950R3

Dear Dr. Fang,

We’re pleased to inform you that your manuscript has been judged scientifically suitable for publication and will be formally accepted for publication once it meets all outstanding technical requirements.

Kind regards,

Floris Vermeulen

Academic Editor

PLOS One
---

## [Editor Report · Acceptance letter]

PONE-D-25-01950R3

PLOS One

Dear Dr. Fang,

I'm pleased to inform you that your manuscript has been deemed suitable for publication in PLOS One. Congratulations! Your manuscript is now being handed over to our production team.

Kind regards,

on behalf of

Dr. Floris Vermeulen

Academic Editor

PLOS One